# Characteristics and Outcomes of Stem Cell Transplant Patients during the COVID-19 Era: A Systematic Review and Meta-Analysis

**DOI:** 10.3390/healthcare12050530

**Published:** 2024-02-23

**Authors:** Mona Kamal, Massimo Baudo, Jacinth Joseph, Yimin Geng, Omnia Mohamed, Mohamed Rahouma, Uri Greenbaum

**Affiliations:** 1Department of Symptom Research, The University of Texas MD Anderson Cancer Center, Houston, TX 77030, USA; 2Department of Cardiac Surgery, Spedali Civili di Brescia, 25123 Brescia, Italy; massimo.baudo@icloud.com; 3Hematology and Medical Oncology, University of Pittsburg Medical Center-Hillman Cancer Center, Altoona, PA 16601, USA; 4Research Medical Library, The University of Texas MD Anderson Cancer Center, Houston, TX 77030, USA; ygeng@mdanderson.org; 5Department of Medical Oncology, NCI, Cairo 11796, Egypt; omnia.korani@gmail.com; 6Surgical Oncology Department, National Cancer Institute, Cairo 12613, Egypt; mmr2011@med.cornell.edu; 7Cardiothoracic Surgery Department, Weill Cornell Medicine, New York, NY 10065, USA; 8Department of Hematology, Soroka University Medical Center, Beer Sheva 8410501, Israel; urigr@clalit.org.il; 9Faculty of Health Sciences, Ben Gurion University of the Negev, Beer Sheva 8410501, Israel

**Keywords:** cancer, COVID-19, meta-analysis, mortality, stem cell transplant

## Abstract

This systematic review and meta-analysis aims to identify the outcomes of stem cell transplant (SCT) patients during the COVID-19 era. Pooled event rates (PER) were calculated, and meta-regression was performed. A random effects model was utilized. In total, 36 eligible studies were included out of 290. The PER of COVID-19-related deaths and COVID-19-related hospital admissions were 21.1% and 55.2%, respectively. The PER of the use of hydroxychloroquine was 53.27%, of the receipt of immunosuppression it was 39.4%, and of the use of antivirals, antibiotics, and steroids it was 71.61%, 37.94%, and 18.46%, respectively. The PER of the time elapsed until COVID-19 infection after SCT of more than 6 months was 85.3%. The PER of fever, respiratory symptoms, and gastrointestinal symptoms were 70.9, 76.1, and 19.3%, respectively. The PER of acute and chronic GvHD were 40.2% and 60.9%, respectively. SCT patients are at a higher risk of severe COVID-19 infection and mortality. The use of dexamethasone improves the survival of hospitalized SCT patients with moderate to severe COVID-19 requiring supplemental oxygen or ventilation. The SCT patient group is a heterogeneous group with varying characteristics. The quality of reporting on these patients when infected with COVID-19 is not uniform and further prospective or registry studies are needed to better guide clinical care in this unique setting.

## 1. Introduction

Preparation for safe effective bone marrow transplantation (BMT) for hematological cancers in the era of COVID-19 starts as early as the pre-transplantation screening phase [1]. During the management of BMT candidates, identification of carriers of COVID-19, and close monitoring of high-risk patients for early diagnosis of infection, with prompt clinical management of highly suspicious cases with false-negative results, are vitally important as part of effective clinical care. In addition, special considerations for infection prevention are raised before, during, and after BMT: psychological aspects, limitations on providers availability, availability of healthy donors, travel to BMT centers, and separation of patients from family, especially in the pediatric population [2,3,4,5,6].

The infectious diseases working parties of the American Society for Transplantation and Cellular Therapy and The European Society for Blood and Marrow Transplantation collaborated to overcome medical challenges during the COVID-19 era and optimize clinical care for stem cell transplantation (SCT) recipients. This much-needed effort encourages transplantation centers to continuously collect patient data and outcomes, initiates educational activities for providers, patients, and caregivers, and issues updated recommendations [7,8,9,10,11,12].

Recently, the number of bone marrow transplants has risen, and their outcomes have improved. This may be attributed to a sustained reduction in the rates of new COVID-19 cases; increased availability of vaccines, personal protective equipment, and trained staff; increased testing capacity; and adherence to social distancing, telemedicine visits, and patient education [2,5,13,14,15,16]. All of these factors also help the performance of BMT in urgent cases by reducing the burden on providers. Despite these efforts, SCT-related clinical care varies across institutions (guided by community prevalence and availability of infrastructure, personal protective equipment, and trained staff). At the early stages of the pandemic, some centers used unapproved agents for COVID-19 treatment based on new data that emerged almost daily, albeit from nonrandomized trials and after short follow-up times [2,12,17]. Additionally, clinical trial recruitment must still be improved as it has been compromised during the COVID-19 pandemic [18,19]. Furthermore, COVID-19 infection manifests in a wide range of symptoms that differ in profile and severity [20] and the published data regarding the clinical care and patients’ characteristics and outcomes during and after BMT are heterogeneous [4,21].

To date, there are no consensus guidelines as to the management of COVID-19 infections in the setting of pre- and post-BMT care. This systematic review and meta-analysis summarizes available data from studies addressing the impact of COVID-19 on BMT recipients to better guide precision care for these patients and overcome challenges in the transplantation setting. Such data are critical for patient prioritization and safety as well as enhancing multidisciplinary teamwork to improve clinical outcomes, reduce the burden on providers, and optimize the selection of alternative care plans and therapies as well as the use of medical resources [22,23].

## 2. Materials and Methods

This systematic review/meta-analysis was conducted according to the preferred reporting items for systematic reviews and meta-analyses guidelines [24].

### 2.1. Literature Search Engines and Terms Used

The MEDLINE (Ovid, New York, NY, USA), Embase (Ovid), Web of Science (Clarivate, London, UK), PubMed, and Cochrane Library (John Wiley & Sons, Hoboken, NJ, USA) databases were searched for publications in the English language from 1 December 2019 to 1 January 2021. The following search terms were used, with subject headings and keywords searched as needed: COVID-19, severe acute respiratory syndrome coronavirus 2, SARS-CoV-2, coronavirus infections, novel coronavirus, cancer, neoplasms, tumor, leukemia, lymphoma, melanoma, carcinoma, sarcoma, oncology, stem cell transplantation, bone marrow transplantation, etc. The search terms were combined with “or” if they represented similar concepts or combined with “and” if they represented different concepts. The complete search strategies are detailed in Appendix A.

### 2.2. Study Types Selected and Inclusion Criteria

The initial search included original studies (case reports, case series, and observational studies) as well as all types of editorials if they included enough patient-related clinical information owing to the rarity of the data. This is due to the rapid reporting during the COVID-19 pandemic, with some authors reporting valuable data in editorials, letters, comments, and conference papers. Eligible studies were required to address the following measurable outcomes in patients with hematological cancers who underwent BMT during the COVID-19 pandemic: rate of COVID-19 infection, emergency room visits, intensive care unit admission, hospital readmission, complications, thromboembolic events, need for supplemental oxygen therapy, need for invasive ventilation, fungal and other opportunistic infections, reinfection with COVID-19, and overall survival. We excluded other types of publications, including review articles, guidelines, experience, consensus, and abstracts. The Joanna Briggs Institute Critical Appraisal tools’ Checklist for Case Reports and Case Series and the Newcastle–Ottawa Quality Assessment Scale for Cohort Studies were used for the critical appraisal and quality assessment of the included papers [13,25,26], Appendix A.

### 2.3. Data Extraction and Statistical Analysis

Excel software (Microsoft, v11.0, Redmond, WA, USA) was used for data extraction. Categorical variables were expressed as frequencies, whereas continuous variables were reported as mean (± SD) values. The following patient variables were extracted from the included articles: age, sex, comorbidities, previous treatment, disease type, type of SCT, timing of SCT, timing of COVID-19 infection, treatment received for COVID-19, treatment delay or interruption owing to COVID-19, presence of graft-versus-host disease (GvHD), type of GvHD (acute or chronic), receipt of immunosuppressive agents (e.g., tacrolimus and sirolimus) at the time of COVID-19 diagnosis, and receipt of steroids at the time of COVID-19 diagnosis. Two investigators performed data extraction independently.

Pooled event rates (PERs) with 95% CIs were calculated. The inverse of the variance in the estimate was estimated using the DerSimonian–Laird method with a random effects model. Studies with double zeros were included in our meta-analysis and treatment arm continuity correction was applied in studies with zero-cell frequencies.

Hypothesis testing for equivalence was set at a two-tailed p-value of 0.05. Determining the heterogeneity was based on Cochran’s Q test with I^2^ values. In the case of heterogeneity (I^2^ > 50%), individual study inference analysis was performed via leave-one-out sensitivity analysis.

Funnel plots by graphical inspection and the Egger regression test were used for the assessment of publication bias. In the case of asymmetry positivity, visual assessment and Duval and Tweedie’s trim-and-fill methods were performed.

All analyses were performed using the R computing language (version 4.1.0) and RStudio application (version 1.4.1717) with the meta and metafor packages.

## 3. Results

A total of 290 studies were identified in the databases. After the exclusion of duplicate articles, 208 studies were screened, excluding 151 studies that were not eligible. This left 57 full-text articles to be assessed for eligibility. Thirty-six of these studies met our eligibility criteria [16,21,27,28,29,30,31,32,33,34,35,36,37,38,39,40,41,42,43,44,45,46,47,48,49,50,51,52,53,54,55,56,57,58,59,60]. These studies included a total of 2011 patients, 516 of whom underwent SCT. Of these patients, 409 were diagnosed with COVID-19 after SCT. Appendix A shows a flow diagram of the preferred reporting items for systematic reviews and meta-analyses guidelines. Table 1 shows the study characteristics and patient demographics. The weighted mean (±SD) age of all SCT patients was 42.1 ± 22.4 years (only 23 articles mentioned patient ages). The weighted percentage of male patients was 61.25% (only 22 articles mentioned patient sex data).

A subgroup analysis was performed to describe the characteristics of the allogenic SCT patients. A mean age of 35.6 ± 23.9 (reported in 15/18 studies) in 104 allogeneic SCT patients was reported; those patients were predominately males (58.0%) (58/100) (reported in 14/18 studies). Matched unrelated, matched related, and haplo-identical (half-matched) donors represented 42.9% (33/77) (reported in 12/18 studies), 38.6% (32/88) (reported in 12/18 studies), and 30.3% (27/89) (reported in 12/18 studies) of allogenic SCT patients in the included studies, respectively. The infection rate in allogenic SCT patients was 93%.

For all SCT patients included in these studies, the COVID-19 infection rate was 79.23%. The mean time from SCT to COVID-19 infection was 10.01 months (95% CI, 8.26–12.14 months; data derived from 24 studies and 333 patients), whereas the range was 6.0 days to 86.4 months. Table 2 shows the outcomes of the meta-analysis. The pooled COVID-19-related mortality was 21.08% (95% CI, 17.18–25.60%) (Figure 1) and that for COVID-19-related hospital admission was 55.20% (95% CI, 47.37–62.78%) (Figure 2). The PER for the use of hydroxychloroquine was 53.27% (95% CI, 36.97–68.91%) (Figure 3). The PER for receipt of immunosuppressive agents was 39.44% (95% CI, 25.11–55.86%) (Figure 4 and Figure 5). Appendix A show the PERs for the use of antivirals (71.61%), antibiotics (37.94%), and steroids (18.46%). As shown in Appendix A, the PERs for the timing of COVID-19 infection varied among all SCT patients diagnosed with COVID-19 within the first 100 days (39%), 100 days to 6 months (34%), and more than 6 months (85%) after SCT. The PER for fever as a presenting symptom was 70.89% (95% CI, 64.22–76.78%). The PER for respiratory symptoms as a presenting symptom was 76.14% (95% CI, 57.31–88.36%). The PER for acute GvHD was 40.20% (95% CI, 11.80–77.16%), whereas that for chronic GvHD was 60.91% (95% CI, 25.39–87.70%).

In a meta-regression analysis (Table 3), we sought to determine whether any of the following factors influenced the risk of COVID-19-related mortality in all included SCT patients: time to COVID-19 infection after SCT, presence of GvHD and its type, use of immunosuppression agents, use of steroids, use of hydroxychloroquine, use of antibiotics, use of antivirals, and hospital admission. We found that none of these factors significantly influenced the rate of COVID-19-related mortality after SCT.

In a meta-regression analysis (Table 3), we sought to determine whether any of the following factors influenced the risk of COVID-19–related mortality in all included SCT patients: time to SARS-CoV-2 infection after SCT, presence of GvHD and its type, use of immunosuppression agents, use of steroids, use of hydroxychloroquine, use of antibiotics, use of antivirals, and hospital admission. We found that none of these factors significantly influenced the rate of COVID-19–related mortality after SCT.

## 4. Discussion

The COVID-19 pandemic has brought challenges to the management of SCT recipients. Transplant recipients are immunocompromised and have an increased risk of severe COVID-19 infection and mortality. Therefore, deciding to proceed with SCT during the pandemic requires careful consideration of the benefits and risks. This meta-analysis provides essential information that can help clinical decisions toward the optimization of SCT recipients’ care during the pandemic. Herein, we summarize the available data from studies that addressed the impact of COVID-19 infection on SCT recipients.

As described above, we found that the PER for COVID-19-related mortality in SCT recipients was 21.08% (95% CI, 17.18–25.60%). This aligns with the attributable COVID-19 mortality rate of 25% in SCT recipients reported by Ljungman et al. [61]. Other researchers found that age greater than 70 years, uncontrolled hematological disease, an Eastern Cooperative Oncology Group grade of 3 or 4, a C-reactive protein level greater than 20 mg/dL, and neutropenia are associated with increased mortality risk [37]; use of azithromycin or low-dose corticosteroids was associated with decreased mortality risk; and use of hydroxychloroquine did not significantly improve the mortality rate. We tested the influence on mortality rate of the time to SARS-CoV-2 infection after SCT; presence of GvHD and its type; use of immunosuppressive agents, steroids, hydroxychloroquine, antibiotics, and antivirals; and hospital admission. None of these factors significantly influenced the COVID-19-related mortality rate after SCT, which may be attributed to small sample sizes in the studies we reviewed.

Nevertheless, the mortality rate in allogeneic SCT recipients who had COVID-19 (32%) at 30 days seemed to be relatively high, even when compared to COVID-19 hospitalized patients in other settings [62] and advanced age, advanced disease stage, and the need for mechanical ventilation were associated with an increased mortality rate [63]. Special safety measures must be implemented when dealing with stem cell donors with COVID-19 [64]. Although a case report demonstrated that transplantation from COVID-19 donors is feasible [65], such strict measures are needed to ensure that the donor is healthy and free of any disease, especially COVID-19, at the time of transplantation. This is important to ensure the safety of the medical team, who closely take care of donors during pretransplant assessment and the donation process. Thus, SCT during the COVID-19 era requires careful consideration of the benefits and risks and patients should be closely monitored for COVID-19 symptoms and managed accordingly.

In terms of the management of COVID-19, hydroxychloroquine, an antimalarial drug, has been administered as an off-label drug. However, its efficacy in treating COVID-19 is controversial and several studies have shown conflicting results, which have drawn attention worldwide [66]. In our meta-analysis, 53% of the included patients received hydroxychloroquine during their COVID-19 treatment, which did not significantly impact mortality. This finding aligns with the results of a study of the effect of hydroxychloroquine versus a placebo in patients with COVID-19, which demonstrated no impact of hydroxychloroquine on the patients’ clinical status, including the mortality rate [67]. Also, according to a Cochrane Library review, treatment with hydroxychloroquine does not alter the need for ventilation in COVID-19 patients or affect their risk of death [68]. Moreover, hydroxychloroquine has been associated with several adverse effects, including cardiotoxicity, and its use in COVID-19 patients may exacerbate underlying cardiac conditions. Therefore, the use of hydroxychloroquine in the treatment of COVID-19 is not currently recommended [67,68].

Optimizing hospital admission of SCT recipients who developed COVID-19 is another critical clinical decision and requires weighing the risks and benefits. Hospital admission is necessary for severe cases of COVID-19 requiring oxygen support or mechanical ventilation. In this meta-analysis, 55.2% of the patients underwent hospital admission during management of COVID-19. Also, El Fakih et al. [69] reported that the admission rate for SCT recipients with COVID-19 was 53%. The decision to admit a COVID-19 patient to the hospital should be based on the severity of the symptoms and the risk factors for severe COVID-19, such as advanced age, the presence of comorbidities, and immunocompromised status. Moreover, hospital admission is necessary for close monitoring of COVID-19 patients at increased risk for serious complications, such as thromboembolism and acute respiratory distress syndrome, especially when they are immunocompromised.

After SCT, immunocompromised patients are at greater risk for a prolonged viral phase than immunocompetent patients are. Some SCT recipients shed infectious viral particles for months after exposure to COVID-19 [49]. The pathogenesis of COVID-19 is characterized by an initial viral phase and may be followed by a severe inflammatory phase. This hyperinflammatory state eventually results in an increased mortality rate [70]. Therefore, treatment of COVID-19 is mainly focused on the control of the initial viral phase and/or the hyperinflammatory response. Antiviral drugs have been used in the treatment of COVID-19 to reduce viral replication and improve clinical outcomes. For example, remdesivir (a nucleoside analog) is an antiviral drug approved by the U.S. Food and Drug Administration for the treatment of COVID-19 in hospitalized patients. A randomized controlled trial compared remdesivir with a placebo in hospitalized patients with COVID-19 and found that the median time to recovery was 10 days in the remdesivir arm and 15 days in the placebo arm [71]. This benefit was also observed in immunosuppressed cancer patients (8%) in that trial, in whom treatment with remdesivir accelerated recovery. The benefit of remdesivir is evident in patients who need low-flow oxygen delivery and those having COVID-19 symptoms for fewer than 10 days, which may be explained by the fact that it supports its effect mainly during the viral phase of COVID-19 infection [71]. Considering the potentially enhanced impact of antiviral treatments on the outcomes of immunocompromised patients, particularly those with prolonged viral phases, in addition to an overall good safety profile, SCT recipients may benefit the most from COVID-19 treatment with remdesivir. In this meta-analysis, 71.61% of the patients received antiviral treatment during their COVID-19 infections and remdesivir was the most used antiviral.

Furthermore, our meta-analysis demonstrated that 39.44% of the SCT recipients infected with COVID-19 received immunosuppressive drugs. Hence, most severe COVID-19 cases are consequences of hyperinflammatory status and the use of systemic immunosuppressive agents in such cases has been an area of intense investigation. Treatment with dexamethasone was found to improve the survival of hospitalized patients with moderate to severe COVID-19 requiring supplemental oxygen or mechanical ventilation [72]. In addition to dexamethasone, adjuvant use of tocilizumab has resulted in improved clinical outcomes, including survival, in COVID-19 patients regardless of the respiratory support needed. Treatment with tocilizumab has been demonstrated to additively decrease the need for mechanical ventilation and to decrease the mortality rate in those not needing invasive mechanical ventilation at the baseline [73].

Our meta-analysis had some limitations. First, the available studies were mostly observational, retrospective, and limited in sample size and heterogeneous in their reporting details and quality. Many of the variables we looked for were missing. This may introduce selection bias and limit the generalizability of our findings, for example, as it is very likely that patients hospitalized for COVID-19 are much more likely to be described in publications, as opposed to those diagnosed in the primary care setting. Second, the management of COVID-19 patients varies across institutions and countries, which may impact outcomes. Third, the COVID-19 pandemic is rapidly evolving and new information on the topic is constantly emerging. Third, we need to emphasize that collecting and analyzing the data for a systematic review, especially if it includes a meta-analysis, takes a while. Therefore, between the search strategy being performed and the final paper being published, time passes and new papers are being published. Lastly, one of the biggest fears after SCT is catching a COVID-19 infection immediately post-graft period where neutropenia is intense. Neutropenia has not been included in this analysis and we found only three studies where COVID-19 occurred within 100 days of transplantation. Furthermore, there is no way to know if the patients were still neutropenic. We studied all other available factors that may have contributed to higher mortality but we did not study neutropenia because such information was not available.

In summary, based on the current available data, the SCT recipients are at increased risk for severe COVID-19 and COVID-19-related mortality. The use of hydroxychloroquine in the treatment of COVID-19 is controversial and is not recommended owing to its limited efficacy and potential adverse effects and our study did not find evidence for its benefit. Hospital admission is necessary for severe cases of COVID-19 and the risk factors associated with mortality include advanced age, male sex, and an increased comorbidity burden. Antiviral drugs such as remdesivir have produced promising results in the treatment of COVID-19 in hospitalized patients.

## 5. Conclusions

The current literature is severely limited and detailed reporting as well as controlled prospective studies are needed to further understand the prognostication and optimal management of COVID-19 patients in the SCT setting.

## Figures and Tables

**Figure 1 healthcare-12-00530-f001:**
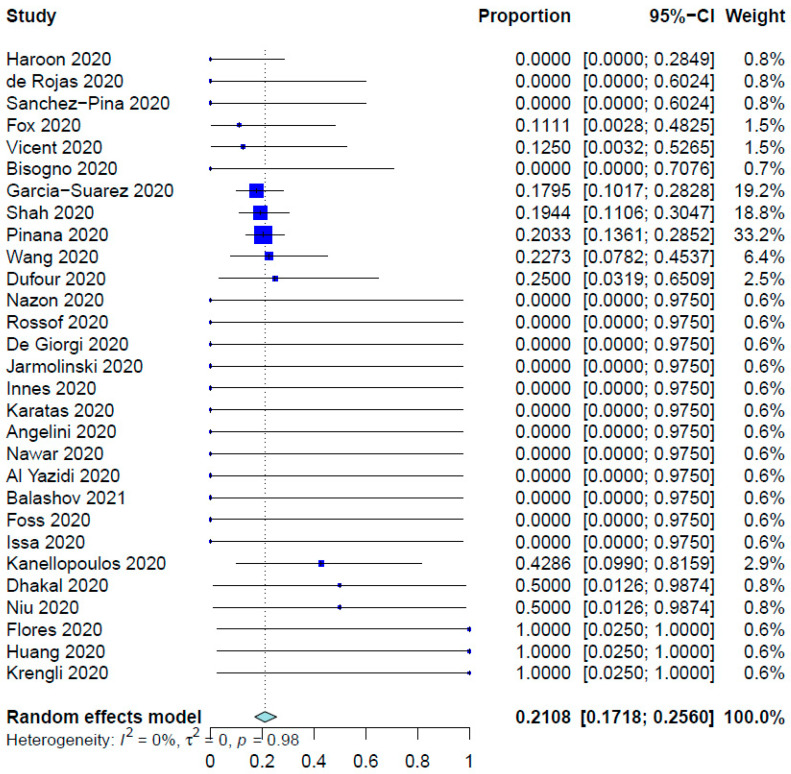
Pooled event rate for COVID-19–related mortality [16,21,27,28,29,30,31,32,34,35,36,37,38,39,40,41,42,43,45,46,47,48,50,52,53,54,56,58,60].

**Figure 2 healthcare-12-00530-f002:**
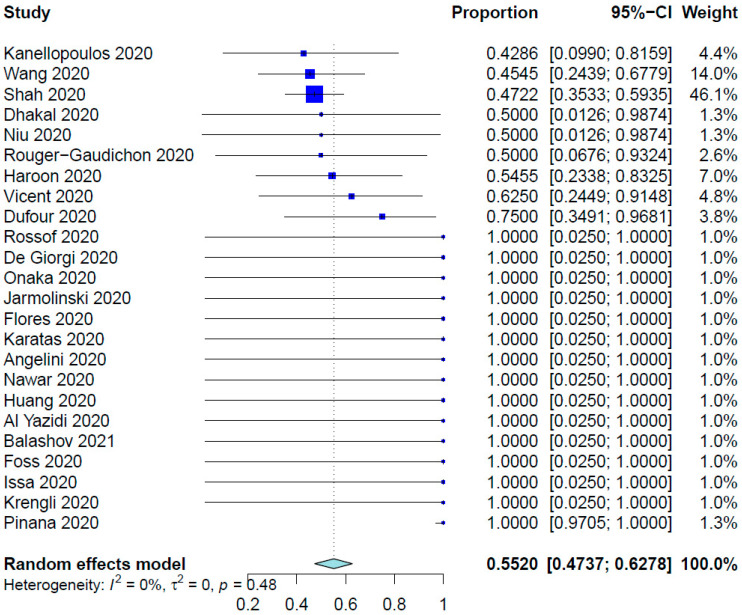
Pooled event rate for COVID-19–related hospital admission [16,21,27,28,30,31,33,34,35,37,38,39,40,41,42,45,46,48,50,52,54,56,58,59].

**Figure 3 healthcare-12-00530-f003:**
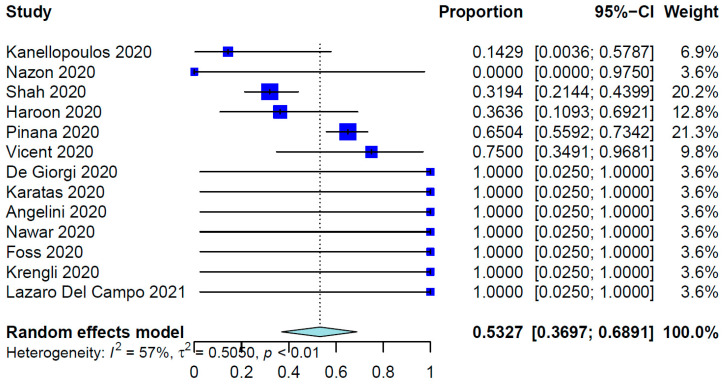
Pooled event rate for the use of hydroxychloroquine [16,28,29,31,37,38,40,41,42,45,52,56,57].

**Figure 4 healthcare-12-00530-f004:**
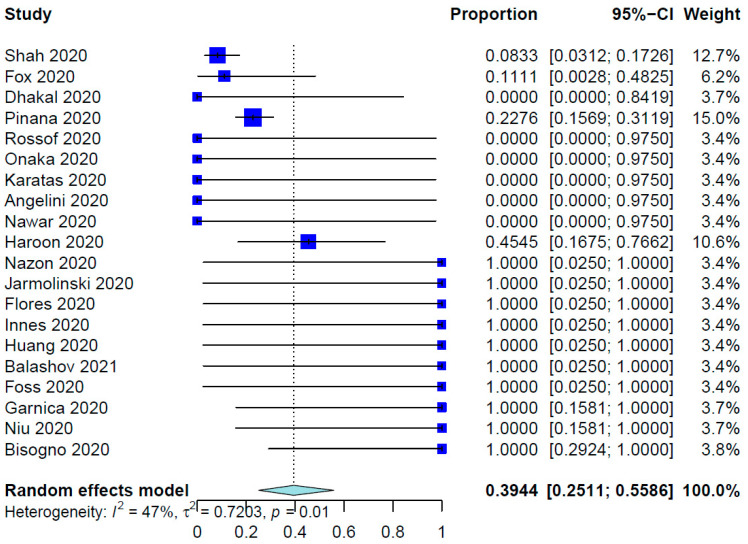
Pooled event rate for receipt of immunosuppressive agents [16,21,29,30,32,33,34,35,36,37,38,39,40,41,42,44,47,50,52,58].

**Figure 5 healthcare-12-00530-f005:**
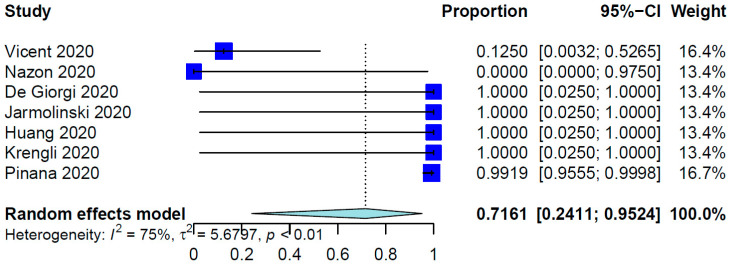
Pooled event rate for the use of antivirals for treatment of COVID-19 [21,28,29,31,34,37,56].

**Table 1 healthcare-12-00530-t001:** The studies reviewed and the respective patients’ demographics.

		*n*						
Author-Year	Type of Article	All Patients	SCT Recipients with COVID-19	Mean Age, Years	Male Patients	Cancer Type	Type of Transplant	Transplant to COVID-19 Infection Time
Dufour 2020 [27]	Letter to the editor	20	8	NA	NA	MM	Auto	7 years
Vicent 2020 [28]	Letter to the editor	8	8	9.25	7	Primary immunodeficiency, MDS, ALL, AML	Allo	1 month–2 years (median, 18 months)
Nazon 2020 [29]	Case report	1	1	17.00	NA	AML-M5	Allo	NA
Rossof 2020 [30]	Case series	6	1	5.00	0	AML	Allo	NA
De Giorgi 2020 [31]	Case report	1	1	33.00	1	Testicular germ cell tumor	Auto	6
Bisogno 2020 [32]	Case series	29	3	NA	NA	JMML, ALL, and HL	Allo (*n* = 3 [100%])	5, 9, and 22 months
Onaka 2020 [33]	Case report	1	1	61.00	1	Transformed follicular lymphoma	Allo	205
Jarmolinski 2020 [34]	Case report	1	1	9.00	0	ALL	Allo	7
Flores 2020 [35]	Letter to the editor	3	1	8.00	0	ALL	Allo	NA
Innes 2020 [36]	Case report	1	1	53.00	1	CML	Allo	3 years
Pinana 2020 [37]	Original article	367	123	54.10	74	AML (23), ALL (13), MDS (10), CMPD (2), NHL (23), CLL (2), plasma cell leukemia (41), aplastic anemia or autoimmune disease (6)	Auto, 58; allo, 65	Auto, 790 (10–10,661); allo, 441 (6–7597)
Karatas 2020 [38]	Case report	1	1	61.00	1	HL, peripheral T-cell lymphoma	Auto	125 days
Dhakal 2020 [39]	Correspondence	7	2	67.60	1	MM	Auto (2)	NA
Angelini 2020 [40]	Case report	1	1	60.00	1	MM	Auto	26 months
Nawar 2020 [41]	Correspondence	3	1	35.00	0	Mediastinal DLBCL	Auto	2 months
Shah 2020 [16]	Original article	77	72	62.00	46	MM (28), NHL (15), leukemia-all types (19), MDS (4), HL (4), amyloidosis (1), MPN (1); SCT recipients	Allo, 35; auto, 37	Median, 782 days
Haroon 2020 [42]	Case series	11	11	40.50	7	MM (3), B-cell ALL (3), HL (1), CML (1), AML (2), DLBCL (1)	Allo, 7; auto, 5	Range, day 5–192 months
de Rojas 2020 [43]	Letter to the editor	15	4	NA	NA	NA	NA	209 days (range, 113–749 days)
Huang 2020 [21]	Case report	2	1	51.00	1	AML	Allo	17 months
Garnica 2020 [44]	Original article	101	2	NA	NA	NA	Allo (2)	>day 100
Kanellopoulos 2020 [45]	Case series	7	7	61.30	2	NA	Auto, 1; allo, 6	7–343 days
Wang 2020 [46]	Original article	58	22	NA	NA	MM	Auto, 22; allo, 0	NA
Fox 2020 [47]	Original article	55	9	NA	NA	NA	Auto, 7; allo, 2	Auto, 0 to >6 months; allo, 0 to <6 months
Al Yazidi 2020 [48]	Letter to the editor	3	1	NA	NA	Primary immunodeficiency	Allo	23 months
Aydillo 2020 [49]	Correspondence	20	18	NA	NA	NA	NA	NA
Balashov 2021 [50]	Original article	1	1	1.00	0	JMML	Allo	3 months (day 99)
Faura 2020 [51]	Letter to the editor	47	8	NA	NA	NA	NA	NA
Foss 2020 [52]	Commentary	1	1	47.00	1	AITL	Allo	17 months
Garcia-Suarez 2020 [53]	Original article	692	78	NA	NA	NA	Auto, 51; allo, 27	8–56 months
Issa 2020 [54]	Letter to the editor	1	1	53.00	1	MCL	Auto	~2 years
Jimenez-Kurlander 2021 [55]	Brief report	321	7	NA	NA	NA	NA	NA
Krengli 2020 [56]	Case report	1	1	62.00	0	MM	Auto	6 months
Lazaro Del Campo 2021 [57]	Expert review	17	1	22.00	10	AML	Allo	NA
Niu 2020 [58]	Correspondence	2	2	60.00	1	B-cell ALL, AML	Allo	5–10 months
Rouger-Gaudichon 2020 [59]	Original article	37	4	NA	NA	NA	NA	NA
Sanchez-Pina 2020 [60]	Original article	92	4	NA	NA	NA	Auto, 3; allo, 1	<6 months

Abbreviations: AITL, Angioimmunoblastic T-cell lymphoma; ALL, acute lymphoblastic leukemia; AML, acute myelogenous leukemia; CLL, chronic lymphocytic leukemia; CML, chronic myelogenous leukemia; CMPD, chronic myeloproliferative disorders; DLBCL, diffuse large B-cell lymphoma; HL, Hodgkin lymphoma; JMML, juvenile myelomonocytic leukemia; MCL, mantle cell lymphoma; MDS, myelodysplastic syndrome; MM, multiple myeloma; MPN, Myeloproliferative neoplasms; NA, not available; NHL, non-Hodgkin lymphoma.

**Table 2 healthcare-12-00530-t002:** (A) Outcomes of meta-analysis for all SCT patients. (B) Outcomes of meta-analysis for allogenic SCT patients.

**(A)**					
**Outcome**	**No. of Studies**	**No. of Patients**	**Effect**	**95% CI**	**Heterogeneity: I^2^, *p*-Value**
COVID-19–related death	29	368	21.08%	17.18–25.60%	0%, 0.9841
Time from SCT					
0–100 days	3	74	38.64%	3.26–92.18%	77.3%, 0.0122
100 days–6 months	4	77	34.19%	6.20–80.34%	69.9%, 0.0189
<6 months	7	151	29.98%	10.48–61.03%	68.1%, 0.0044
>6 months	13	88	85.32%	75.54–91.62%	0%, 0.8550
GvHD	8	20	59.36%	34.21–80.41%	11.6%, 0.3397
Acute	5	16	40.20%	11.80–77.16%	36.6%, 0.1775
Chronic	6	18	60.91%	25.39–87.70%	39.4%, 0.1427
Immunosuppression	20	236	39.44%	25.11–55.86%	46.7%, 0.0116
Use of steroids	12	223	18.46%	6.47–42.55%	53.2%, 0.0149
Use of hydroxychloroquine	13	229	53.27%	36.97–68.91%	56.9%, 0.0058
Use of antibiotics	12	100	37.94%	25.89–51.68%	9.8%, 0.3494
Use of antivirals	7	136	71.61%	24.11–95.24%	74.5%, 0.0006
Symptoms					
Fever	8	201	70.89%	64.22–76.78%	0%, 0.6860
Respiratory	11	205	76.14%	57.13–88.36%	53.9%, 0.0168
Gastrointestinal	3	197	19.25%	12.47–28.52%	0%, 0.4789
Hospital admission	24	273	55.20%	47.37–62.78%	94.8%, <0.0001
**(B)**					
**Outcome**	**No. of Studies**	**No. of Patients**	**Effect**	**95% CI**	**Heterogeneity: I^2^, *p*-Value**
COVID-related death	15	100	21.73%	14.7–30.91%	0%, *p* = 0.9387
Mean time from SCT to COVID infection (month)	13	98	9.99	7.79–12.81%	100%, *p* < 0.0001
GvHD	11	24	57.56%	38.62–74.52%	0%, *p* = 0.8550
Acute	8	16	42.03%	22.21–64.80%	0%, *p* = 0.7675
Chronic	8	16	53.95%	26.86–78.89%	10.1%, *p* = 0.3519
Immunosuppression	14	88	51.27%	40.93–61.50%	0%, *p* = 0.6718
Use of steroids	11	89	26.16%	11.92–48.12%	23.6%, *p* = 0.2188
Use of HQ	7	89	45.02%	34.83–55.65%	0%, *p* = 0.4385
Use of antibiotics	8	21	47.75%	28.32–67.88%	0%; *p* = 0.7058
Use of antiviral	5	76	40.71%	30.10–52.25%	0%, *p* = 0.4058
Symptoms					
Fever	5	69	64.25%	52.64–74.40%	0%, *p* = 0.9797
Respiratory	6	70	80.58%	70.04–88.05%	0%, *p* = 0.9987
Hospital admission	12	90	67.18%	57.24–75.79%	0%, *p* = 0.9999

**Table 3 healthcare-12-00530-t003:** Meta-regression analysis of COVID-19–related mortality.

**(A) Meta-regression analysis of COVID-19–related mortality in all SCT patients.**		
**Variable**	**β (± SE)**	***p*-value**
Time from SCT		
0–100 days	0.0035 ± 0.0128	0.7866
100 days-6 months	0.0026 ± 0.0126	0.8360
<6 months	0.0035 ± 0.0101	0.7276
>6 months	0.0574 ± 0.0465	0.2169
GvHD	0.0283 ± 0.0197	0.1507
Acute	0.0031 ± 0.0144	0.8306
Chronic	0.0272 ± 0.0187	0.1450
Immunosuppression	0.0074 ± 0.0063	0.2434
Use of steroids	0.0033 ± 0.0103	0.7482
Use of hydroxychloroquine	0.0003 ± 0.0077	0.9698
Use of antibiotics	0.0116 ± 0.0084	0.1682
Use of antivirals	0.0045 ± 0.0101	0.6568
Hospital admission	0.0005 ± 0.0058	0.9376
**(B) Meta-regression analysis of COVID-19–related mortality in allogenic SCT patients.**		
**Variable**	**β ± SE**	***p*-value**
GvHD total	0.0097 ± 0.0136	0.4738
Acute GvHD	−0.0043 ± 0.0142	0.7628
Chronic GvHD	0.0106 ± 0.0127	0.4057
Immunosuppression	0.0109 ± 0.0100	0.2759
Use of steroids	0.0023 ± 0.0107	0.8301
Use of HQ	−0.0024 ± 0.0172	0.8878
Use of antibiotics	0.0189 ± 0.0135	0.1602
Use of antiviral	0.0178 ± 0.0159	0.2627
Hospital admission	0.0245 ± 0.0193	0.2049

## Data Availability

Data are contained within the article.

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
