# Peer review of "Characteristics and Outcomes of Stem Cell Transplant Patients during the COVID-19 Era: A Systematic Review and Meta-Analysis"

_healthcare, 2024, doi:10.3390/healthcare12050530_

Round 1

Reviewer 1 Report

Comments and Suggestions for Authors

Summary:

1. Homogenize the term COVID-19 throughout the manuscript.

2. Specify which study designs were considered.

Introduction:

3. The introduction is clear, however, it could benefit from specifying the knowledge gap and importance of the work.

Materials and methods:

4. The type of studies that should be included in the search strategy has not been defined. What's more, the search strategy is not clear.

5. The selection of evidence is very heterogeneous, is it based on observational studies? Have the selected letters to the editor been considered observational studies?

Conclusions:

6. The conclusions must be in accordance with the objectives set out in the study

Comments on the Quality of English Language

Minor editing of English language required

Author Response

RE: Manuscript ID: healthcare-2856189

We respectfully resubmit our manuscript, “Characteristics and outcomes of stem cell transplant patients during COVID-19 era: a systematic review and meta-analysis,” as a systematic review and meta-analysis for consideration to be published in Healthcare. Our revisions in response to the reviewers’ comments make the manuscript clearer. We are summarizing the changes we made in response to reviewers’ comments. 

Reviewer 1

  1. Homogenize the term COVID-19 throughout the manuscript.

Thank you for the comment. The term COVID-19 has been homogenized thought the manuscript to be “COVID-19”

  1. Specify which study designs were considered.

We appreciate the reviewer’s question. The initial search included original studies (case reports, case series, and observational studies) and all types of editorials if they included enough patient-related clinical information owing to the rarity of the data. We also mentioned excluding other types of publications, including review articles, guidelines, experience, consensus, and abstracts. All this information and the study designs are already mentioned under the “Study selection and inclusion criteria” section. Kindly also see more details in Table 1 describing the type of each included study. 

  1. The introduction is clear; however, it could benefit from specifying the knowledge gap and importance of the work.

Thank you for the comment. The knowledge gap we tried to fill is that the published data regarding clinical care and patients' characteristics and outcomes during and after BMT are heterogeneous. Thus, we designed this review to summarize available data from studies addressing the impact of COVID-19 on BMT recipients to better guide precision care. Per the reviewer's suggestion, we added the following sentence to clarify the knowledge gap addressed by this study:" To date, there are no consensus guidelines as to the management of COVID-19 infections in the setting of pre and post-BMT care. "

  1. The type of studies that should be included in the search strategy has not been defined. What's more, the search strategy is not clear.

We appreciate the chance to clarify this point. The detailed literature search strategy has been provided in supplementary tables S1-S4. The search strategy didn't restrict the search results by type of study. Eligible studies were selected from all search results described in the "Study Selection and Inclusion Criteria" section. Under this section, we also mentioned that we excluded other types of publications, including review articles, guidelines, experience, consensus, and abstracts. To clarify this, as the reviewer requested,  we changed the headlines in the methods section (sections 2.1 and 2.2)

  1. The selection of evidence is very heterogeneous, is it based on observational studies? Have the selected letters to the editor been considered observational studies?

Thank you for the chance to clarify this one point again. Although the initial search included original studies (case reports, case series, and observational studies) as well as all types of editorials if they included enough patient-related clinical information owing to the rarity of the data, we did not include review articles, guidelines, experience, consensus, and abstracts. We added a reference to the limitation of the quality of selected studies in the limitations section of the discussion (lines 268-273).

  1. The conclusions must be in accordance with the objectives set out in the study.

Thank you for the review’s comment. We have made the conclusion more reflective in accordance with the study objectives: "Current literature is severely limited, and detailed reporting, as well as controlled prospective studies, are needed to further understand the prognostication and optimal management of COVID-19 patients in the SCT setting.”

Reviewer 2 Report

Comments and Suggestions for Authors

This is a systematic review and meta-analysis of clinical characteristics and especially outcomes in patients undergoing hematopoietic stem cell transplants with COVID-19. The subject is very interesting and brings together the published works, providing important information for this severely immunocompromised population.

As this is a meta-analysis, I don't have many considerations in the methodology, but I would like the authors to clarify some topics:

1) The year of the search was until 2021 and we know that in 2022 the Covid-19 pandemic was still having a great impact on hospitals and the population with other diseases. I consider that it will not be possible to redo the meta analysis, but it is possible to put this in the discussion, as certainly, more articles were published in 2022.

2) In lines 94 to 99, the authors consider that for the study to be eligible, they should have different variables, however, many of these variables were not mentioned in the results, such as: intensive care unit admission, hospital readmission, complications, thromboembolic events, need for supplemental oxygen therapy, need for invasive ventilation, fungal and other opportunistic infections, reinfection with SARS-CoV-2. Please clarify.

3) In the same way, the main outcome that the article should have in order to be eligible is mortality. The authors selected 36 articles, but only 29 had the mortality outcome. Please clarify

In the results:

1) Table 1 lists the 36 selected articles, but not all of them are listed in the references (Karatas, Garnica, etc...)

In the discussion:

1) One of the biggest fears in this scenario is transplanting patients and they acquire Covid-19 in the immediate post-graft period where neutropenia is intense. Neutropenia has not been studied, and the authors have only 3 studies where COVID-19 occurred within 100 days of transplantation. Even so, we have no way of knowing if the patient was still neutropenic. The authors studied some factors that may have contributed to higher mortality, but they did not study neutropenia. Please put this item in the discussion.

2) The mortality found in the review was 21% and the mortality found in reference 29 was 34%. Can we consider mortality high in the meta-analysis? Is it not worth comparing with the mortality of non-transplanted COVID-19 patients for this statement?

Author Response

RE: Manuscript ID: healthcare-2856189

We respectfully resubmitting our manuscript, entitled “Characteristics and outcomes of stem cell transplant patients during COVID-19 era: a systematic review and meta-analysis” as a systematic review and meta-analysis for consideration to be published in Healthcare. We believe that such the revisions we made in response to the reviewers’ comments make the manuscript clearer. We are providing a summary of the changes we made in response to reviewers’ comments. 

Reviewer 2

This is a systematic review and meta-analysis of clinical characteristics and especially outcomes in patients undergoing hematopoietic stem cell transplants with COVID-19. The subject is very interesting and brings together the published works, providing important information for this severely immunocompromised population.

As this is a meta-analysis, I don't have many considerations in the methodology, but I would like the authors to clarify some topics:

  • The year of the search was until 2021 and we know that in 2022 the Covid-19 pandemic was still having a great impact on hospitals and the population with other diseases. I consider that it will not be possible to redo the meta-analysis, but it is possible to put this in the discussion, as certainly, more articles were published in 2022.

We agree with the reviewer that we have to address this point. As you know it takes a while to collect and analyze the data for a systematic review, especially if it includes a meta-analysis. Therefore, between the search strategy and the final paper being published, time passes, and new papers are being published. We added that to the discussion as advised by the reviewer. (lines 276-279).

  • In lines 94 to 99, the authors consider that for the study to be eligible, they should have different variables, however, many of these variables were not mentioned in the results, such as: intensive care unit admission, hospital readmission, complications, thromboembolic events, need for supplemental oxygen therapy, need for invasive ventilation, fungal and other opportunistic infections, reinfection with SARS-CoV-2. Please clarify.

Thank you for the reviewer’s comment. The intent was to retrieve all such information. Unfortunately, the information was rarely provided. As such, analysis was not performed for those variables. However, we included all information in the analysis whenever available. Kindly check the supplementary materials for more information about such data.  We also added a sentence referring to this as a limitation of our study in the limitation section of the discussion (lines 270-273).

  • In the same way, the main outcome that the article should have in order to be eligible is mortality. The authors selected 36 articles, but only 29 had the mortality outcome. Please clarify.

We appreciate the chance to clarify this point. Due to the limited available information, all papers were considered, irrespective of mortality outcomes. We updated the text to clarify this point. Under the methodology section, we mentioned “Eligible studies were required to address the following measurable outcomes in patients with hematological cancers who underwent BMT during the COVID-19 pandemic: rate of COVID-19 infection, emergency room visits, intensive care unit admission, hospital readmission, complications, thromboembolic events, need for supplemental oxygen therapy, need for invasive ventilation, fungal and other opportunistic infections, reinfection with COVID-19SARS-CoV-2, and overall survival. As mentioned above, we also addressed this in the limitations section.

In the results: 1) Table 1 lists the 36 selected articles, but not all of them are listed in the references (Karatas, Garnica, etc...)

Thank you for the comment. We have revised the references and cited all included papers in the first paragraph of methodology.

In the discussion: 1) One of the biggest fears in this scenario is transplanting patients and they acquire Covid-19 in the immediate post-graft period where neutropenia is intense. Neutropenia has not been studied, and the authors have only 3 studies where COVID-19 occurred within 100 days of transplantation. Even so, we have no way of knowing if the patient was still neutropenic. The authors studied some factors that may have contributed to higher mortality, but they did not study neutropenia. Please put this item in the discussion.

Thank you for bringing up this important point. Sadly, data on neutropenia was not available. As advised, we added that to the discussion (lines 280-285). “One of the biggest fears after SCT is catching COVID-19 infection immediately post-graft period where neutropenia is intense. Neutropenia has not been included in this analysis, and we found only three studies where COVID-19 occurred within 100 days of transplantation. Further, there is no way to know if the patients were still neutropenic. We studied all other available factors that may have contributed to higher mortality, but we did not study neutropenia because such information was unavailable”.

2) The mortality found in the review was 21%, and the mortality found in reference 29 was 32%. Can we consider mortality high in the meta-analysis? Is it not worth comparing with the mortality of non-transplanted COVID-19 patients for this statement?

Thank you for the reviewer’s comment. Reference 29 was about the mortality after allogeneic SCT at 30 days. Comparing the mortality after SCT with the mortality of non-transplanted patients is out of the scope of this review. However, we emphasize that special considerations must be implemented while treating SCT patients during COVID-19. To address the reviewer's comment, we added to lines 197-198 " …seemed to be relatively high, even when compared to COVID-19 hospitalized patients in other settings." and added a reference, https://www.ncbi.nlm.nih.gov/pmc/articles/PMC8025591/ that address the mortality in hospitalized patients during COVID-19 era.

Round 2

Reviewer 1 Report

Comments and Suggestions for Authors

The authors have answered satisfactorily

Comments on the Quality of English Language

Minor editing of English language required